# Combined Rehabilitation Protocol in the Treatment of Osteoarthritis of the Knee: Comparative Study of Extremely Low-Frequency Magnetic Fields and Soft Elastic Knee Brace Effect

**DOI:** 10.3390/healthcare11091221

**Published:** 2023-04-25

**Authors:** Teresa Paolucci, Daniele Porto, Raffaello Pellegrino, Ornela Sina, Andi Fero, Sara D’Astolfo, Sara Franceschelli, Antonia Patruno, Augusto Fusco, Mirko Pesce

**Affiliations:** 1Department of Oral, Medical and Biotechnological Sciences, Physical Medicine and Rehabilitation, University of G. D’Annunzio Chieti-Pescara, 66100 Chieti, Italy; 2Unit of Physical Medicine and Rehabilitation, Don Orione Institute, 65128 Pescara, Italy; 3Department of Scientific Research, Campus Ludes, Semmelweis University, 6912 Lugano, Switzerland; 4Department of Medicine and Aging Science, University of G. D’Annunzio Chieti-Pescara, 66100 Chieti, Italy; 5UOC Neuroriabilitazione ad Alta Intensità, Fondazione Policlinico Universitario A. Gemelli IRCCS, 00168 Rome, Italy

**Keywords:** knee, osteoarthritis, sleeve, rehabilitation, magnetic field, soft brace

## Abstract

The investigation of this observational case–control study aimed at determining the effectiveness of a combined treatment of extremely low-frequency electromagnetic fields (ELF) with a soft elastic knee brace versus ELF alone in knee osteoarthritis (KOA) with respect to a reduction in pain and functional recovery. We hypothesized that the combined use of ELF and a soft elastic knee brace may provide better results. Thirty-five patients (N = 35, divided into Group 1 = ELF and Group 2 = ELF with the soft elastic knee brace) were analyzed. The rehabilitative protocol consisted of 10 sessions of antiphlogistic and antiedema programs (first cycle) for 2 weeks, followed by twelve sessions of bone repair and connective tissue repair programs (second cycle) in patients with knee osteoarthritis (KOA) for 4 weeks. Patient evaluations were conducted at baseline (T0) and after 2 (T1) and 4 (T2) weeks of treatment. A follow-up evaluation was conducted 6 weeks after treatment (T3). The LIMFA© Therapy System was used to create multifrequency magnetoelectric fields with an intensity of 100 µT and a low-frequency field. The Incrediwear Cred 40 knee sleeve (Incred) was used for alleviating knee pain. The Visual Analogue Scale (VAS), the Knee Injury and Osteoarthritis Outcome Score (KOOS), and the Lysholm score (Ls) were used as outcome measures. The results showed that pain at rest (Vr), pain in motion (Vm), KOOS, and Ls were significantly affected by ELF over time. In conclusion, Group 2 had a better response in terms of pain resolution at rest (*p* < 0.05) and a concurrent better response at T3 in terms of functional recovery (*p* < 0.05).

## 1. Introduction

Osteoarthritis (OA) is a chronic degenerative joint disease that occurs most commonly in people over 45 years of age. It is characterized by articular cartilage loss, synovial inflammation, and the remodeling of subchondral bone. OA has been shown to be associated with joint pain, stiffness, loss of function, reduced quality of life (QOL), and mortality. The treatment of OA traditionally comprises nonpharmacological and pharmacological management; however, if symptoms persist, surgery may be considered. Current treatments are limited by small effect sizes and adverse side effects. In recent years, there has been much emphasis on nonpharmacological management such as education, physiotherapy, and exercise therapy to relieve symptoms and improve function in those with OA [1].

Magnetotherapy provides a non-invasive, safe, and easy method to directly treat the site of injury, the source of pain and inflammation, and it is widely used in rehabilitation in OA [2,3]. Briefly, a pulsed electromagnetic field (PEMF) can promote the proliferation of osteoblasts when its frequency is 7.5–15 Hz or 50–75 Hz and the intensity is 0.4–1.5 mT or 3.8–4.0 mT [3]. This underlines EMFs with different frequencies and intensities as able to exert distinct bioeffects on specific bone cells with good results for the reduction in pain and improvement of function in knee osteoarthritis (KOA) [4,5]. KOA is believed to be highly prevalent today because of recent increases in life expectancy and body mass index (BMI) and remains the most challenging arthritic disorder, with a high burden of disease and no available disease-modifying treatment [6,7]. It is estimated that 32.5 million US adults have clinical OA, with the most common sites being the knee and hip [8]. Thus, a multidisciplinary and sustained international effort involving all major stakeholders is required.

There are different rehabilitative electromagnetic field programs used in KOA based on short protocols, such as the program by Nelson et al., which consists of 2 weeks of treatment (15 min per session, twice daily) with 6.8 MHz and an intensity of 30 Gauss [5], or the longer program by Bagnato et al. that proposed a 12 h/daily treatment for 1 month. Both approaches proved effective and well tolerated by the patient [9]. Ay et al. applied PEMFs for 30 min, 5 times/week for 3 weeks on KOA with good results for pain reduction and functional recovery [10,11]. Additionally, Özgüçlü et al. successfully used PEMFs in patients with knee pain using a protocol lasting 30 min per session [12,13].

The difficulty often encountered in rehabilitation is the choice of a unique and shared physical therapy magnetic fields protocol in the treatment of symptomatic KOA. Therefore, the choice is entrusted to the physiatrist and physical therapists based on the characteristics and needs of the patient. Surely, short protocols allow better compliance by the patient [14] and an earlier start to the therapeutic rehabilitation exercise, with a reduced use of pain-relieving drugs. Containing and adequately managing pain in KOA allows physical therapists to face it and adequately guide their clinical decision-making by summarizing the safest and most efficacious exercise options [15]. Patient education, physical exercise, and weight loss may constitute the first-line KOA treatment approach.

To reduce pain, improve physical function, and, possibly, slow disease progression in KOA, the use of knee braces has often been suggested [16]. These are generally the main purposes of knee braces [17], but the optimal choice for an orthosis remains unclear, and long-term implications are lacking. A variety of different bracing types, manufacturers, and products are currently available on the market. Short-lever elastic knee braces have been used to improve pain, specifically during squats or walking, and daily use or the use of soft knee braces while resting is suggested to provide moderate pain relief and small-to-moderate effects on performance-based physical function in patients. Several authors, with respect to these findings, highlight the importance of soft braces to improve pain reduction and physical function in both the short and long term in KOA treatment, but additional high-quality studies are warranted to improve confidence in the findings [18,19,20,21].

Considering these premises and the evidence of the literature, we hypothesized that the combined use of magnetic fields and a soft elastic knee brace may have a better response in the resolution of pain in KOA patients compared to magnetic fields alone.

The aim of the study was to evaluate the effect of the combined treatment of ELF and the soft elastic knee brace compared to ELF alone, with pain reduction as the primary outcome and functional recovery as the secondary outcome.

## 2. Materials and Methods

### 2.1. Study Design and Population

An observational case–control study was conducted per the Strobe Guidelines [22] to determine the combined effect of ELF and soft elastic Knee brace vs. ELF alone in treating acute painful knee osteroarthritis.

Patients with acute pain in KOA were screened at the physical medicine and rehabilitation outpatient clinic of Don Orione Institute of Pescara (Italy) from December 2020 to February 2022 by a physician skilled in physical and rehabilitation medicine. 

The study was brought to the attention of the Department of Oral Medical Sciences and Biotechnologies of the G. D’Annunzio University of Chieti (Italy) and was approved by the National Ethics Committee of the Don Orione Board. Informed consent was obtained from all individual patients included in the study. The patients were treated in accordance with the World Medical Association Declaration of Helsinki.

All procedures performed in this study were in accordance with the ethical standards of the institutional and/or national research committee and with the 1964 Declaration of Helsinki and its later amendments or comparable ethical standards.

### 2.2. Inclusion and Exclusion Criteria

The inclusion criteria were age between 40 and 70 years, diagnosis of acute pain in KOS < than 1 month, VAS > 3 according to the American College of Rheumatology Criteria [23], and stage II classification per the Kellgren–Lawrence classification (KLc) [24]. The exclusion criteria were red flags (such as cancer, hematological diseases, and specific rheumatological pathologies) [25], bleeding disorder, local infection, pregnancy, patient refusal or noncompliance, pacemaker use, candidacy for knee joint replacement or any intra-articular injection within six months, addiction to opioid drugs, thrombocytopenia, use of anticoagulant or antiaggregant, and recent myocardial infarction or stroke. We excluded patients who were undergoing any other type of physiotherapy or conservative treatments during the study period. KOS was diagnosed based on the clinical examination and radiographic images (X-ray) in the standing position. 

Finally, patients were included consecutively and formed two groups that are indicated as LIMFA therapy without (Group 1) or with (Group 2) Incred.

### 2.3. Outcome Measures

The Visual Analogue Scale (VAS) was used to measure knee pain outcomes: patients were asked to mark the point that corresponded to their perceived pain intensity on a 10 cm line, with 0 indicating the absence of pain and 10 reflecting the most severe pain [23]. The patient was asked to indicate both pain at rest (Vr) and pain in motion (Vm).

The Knee Injury and Osteoarthritis Outcome Score (KOOS) is a questionnaire designed to assess short-term and long-term patient-relevant outcomes following a knee injury or OA [26,27]. The KOOS is self-administered and assesses 5 outcomes: pain, symptoms, activities of daily living (ADL), sport and recreation function, and knee-related quality of life. This scale was used to evaluate pain and function after patients reported the pain and ADL domains.

Lysholm score (Ls), which measures the ability to manage in everyday life [28,29], was used as a patient-reported outcome measure questionnaire (PROM). The total score varies from a minimum of 0 to a maximum of 100, where 0 corresponds to complete disability, while 100 corresponds to no symptoms. The total scores are divided into the following categories: poor: <65 points; right: 65–83 points; good: 84–90 points; excellent: >90 points.

### 2.4. Magnetic Fields Protocol

The LIMFA © Therapy System (Eywa srl; Rimini, Italy) (ISO9001 certification number 390263—commercially available) was used to create multifrequency magnetoelectric fields with an intensity of 100 µT and a low-frequency field. 

The Limfa’s patented ELF pre-set sequences are complex magnetic fields, pulsed and bipolar, that target specific ion channels in order to obtain the desired biological reaction. Depending on the ion channel, the frequencies vary from 2 to 80 Hz with a variable field intensity (lower than 100 µT) and variable waveform. The abovementioned parameters are not random but are pre-defined for each stimulation program.

Patients participated in ten (n = 10) sessions of antiphlogistic and antiedema programs (first cycle) (15 min each) for 2 weeks, 5 days/week, followed by, without interruption, twelve (n = 12) sessions (20 min each) of bone repair and connective tissue repair programs (second cycle) for 4 weeks, 3 days/week (Figure 1 and Figure 2).

### 2.5. Soft Knee Brace

The Incrediwear Cred 40 knee sleeve (Incred) [30] is embedded with carbonized charcoal and germanium, which is a nontoxic semiconductor metalloid located between tin and silicone in the periodic table. The resistance of germanium decreases, while the temperature of semiconductors increases with more “free” electrons, allowing for higher conductivity (Figure 3). 

### 2.6. Data Collection

The questionnaires were given to patients when they first received the knee sleeve, before starting magnetic field therapy (T0 = at baseline). They were also given to patients at the end of the first cycle of magnetotherapy (T1 = after 2 weeks), at the end of the second cycle of magnetotherapy (T2 = after 4 weeks), and 6 weeks after the end of all treatment (T3 = follow-up). Each patient was observed and received the treatment for a total of three months. 

### 2.7. Sample Size

The sample size was calculated, considering pain intensity per the VAS as the primary outcome. A power analysis of 90% and an alpha level = 0.05 was considered with a difference of 2 points (cm), assuming a standard deviation of ±1.5 for the VAS in relation to the minimal clinically important difference between the groups after treatment. Thus, the analysis would yield a minimum of 16 patients per group, and, based on a 20% potential dropout rate, we included 22 patients per group [31].

### 2.8. Statistical Analysis

Data were checked for normality and homogeneity of variance, which were analyzed with parametric and non-parametric data, respectively. The outcome scales for Vm, Vr, KOOS, KOOS-functional, and Ls differences during treatment, both for those who used and did not use Incred, were analyzed by two-way ANOVA for repeated measures (1 between factor, patients that used Incred or not; 1 within the experimental time points considered) followed by the Bonferroni post hoc test.

Statistical analyses were performed using SPSS 22.0 statistic (SPSS Inc., Chicago, IL, USA) for Windows (IBM). Results are described as means ± SD for each assessment. The level of a statistically significant difference was defined as *p* < 0.05.

## 3. Results

Out of the thirty-five patients (N = 35 divided into Group 1, N = 18, and Group 2, N = 17) (Table 1)), sixteen patients (N = 16) were excluded following the initial observation (N = 51) and nine patients (N = 9) did not join the study. Three patients were excluded from Group 1 because they did not complete the rehabilitation treatment for personal and work reasons. Furthermore, four patients were excluded in Group 2: two because they did not find the brace comfortable and it was not worn according to the indications; two because they did not complete the rehabilitative treatment according to the protocol, taking advantage of recovery sessions for work reasons that lengthened the planned times.

The comparisons of values obtained at different time points (T0–T3) showed that Vr (F_7, 245_ = 35.41, *p* < 0.001), Vm (F_7, 245_ = 74.22, *p* < 0.001), KOOS (F_7, 245_ = 63.29, *p* < 0.001), and Ls (F_7, 245_ = 72.23, *p* < 0.001) were significantly affected by the ELFs over time (Figure 4). No differences were obtained for the KOOS-functional variable.

Patients wore the brace during rest and daily activity from at least 6 h to 8 h a day, every day, for 6 consecutive weeks and during magnetic field therapy.

Notably, the differences recorded for Vr were significantly regulated by Incred, suggesting that this device ameliorates the effect of magnetic treatment at rest (Table 2).

## 4. Discussion

Considering the aim of the study, the results appeared interesting. Above all, compared to the primary outcome of pain reduction, Group 2, who carried out the two treatments in combination, showed a more rapid reduction in knee pain and a better response at rest, which were well maintained at follow-up (*p* < 0.01), with a significantly better functional recovery at T2 (the end of the second rehabilitative cycle) for KOOS, as reported in Table 2 (*p* for VAS at rest = 0.011).

The rehabilitative treatment options in KOA have been widely implemented and researched, but the optimum treatment or combinations of treatment remain unclear [32,33]. Particularly in the early stages of KOA progression, a conservative approach is commonly favored: interventions can be stratified according to the duration of the treatment protocol and outcomes into short term (4–12 weeks), medium term (12–26 weeks), and long term (>26 weeks). The TENS and PEMF showed a positive effect on short-term outcomes for pain. For medium-term outcomes, beneficial effects included weight loss for general pain and function, intra-articular infiltrations (corticosteroids, glucosamine, chondroitin sulfate, and platelet-rich plasma), quality of life, and pain during exercise programs [32].

The ELF protocol adopted and proposed in our study would seem to confirm what was reported by the literature according to short-term benefits with respect to pain in KOA and functional recovery. Our protocol envisaged six weeks of treatment, including a more intensive first phase aimed at reducing pain and a more extensive second phase aimed at functional recovery, a regenerative component, and a further observation period of another six weeks. As suggested by the literature, in the second phase of our treatment, the frequency used was maintained at <50 Hz along with an intensity of 0.1 mT: sinusoidal EMF, with 0.9–4.8 mT and 45–60 Hz, and a static magnetic field with 0.1–0.4 mT or 400 mT, could promote a regenerative response in the bone with osteoblast differentiation and maturation [3]. Instead, in the first phase of our protocol, the frequency was >50, but <100 Hz to have a mainly analgesic action, keeping the intensity of the magnetic field unchanged.

The rationale of the PEMF application to OA resides in its ability to modulate the expression of several elements in cells residing in the joint tissues. Previous in vitro studies have demonstrated that PEMFs enhance the expression of adenosine receptors in chondrocytes and synoviocytes. These receptors play a relevant role in the control of nociception and inflammation, suggesting a possible molecular mechanism for the observed pain reduction. PEMFs have also been proposed as capable of cartilage regeneration. Indeed, biophysical treatment actions are related to a pleiotropic effect on several targets, comprising integrins, ion channels, growth factors, and intracellular pathways, which may contribute to the proliferation and maturation of tissue resident cells, including chondrocytes, possibly counteracting the progressive loss of tissue characterizing degenerative disorders, such as OA. In general, the emission of PEMFs was able to exert a protective action and stimulate cartilage tissue [34,35,36], but there is a lack of defined and reproducible protocols for frequency, intensity, and duty cycle and the possibility of identifying, *in vivo*, a minimum dose effect of the magnetic field necessary for the correct therapeutic response. Viganò M. et al. underlined that PEMFs effectively relieve KOA symptoms in the short term, but they are not superior to other conservative therapies such as physiotherapy [37].

With respect to magnetic field therapy, the best physical parameters to be used in osteoarthritis treatment, such as frequency, are debated in the literature. Negm et al. suggested that a low frequency (≤100 Hz) pulsed sub-sensory threshold electrical stimulation produced is effective in improving physical function but not pain intensity at treatment completion in adults with KOA [38]. Similarly, the same authors described how the heterogeneity of the different protocols was not a significant problem for pain or physical function outcomes and that the effect is similar regardless of the type of pulsed sub-sensory threshold electrical stimulation (frequency ranging from 5 to 100 Hz) and the length of treatment (<12 and ≥12 weeks) [38]. Meanwhile, a short PEMF treatment duration (within 30 min) may achieve favorable efficacy [39,40], but the effects of ELF-EMF depend on their respective codes as frequency, intensity, and waveform [41].

In terms of the length of the treatment period, we have adopted a protocol similar to that used by Thamsborg and colleagues [42], reporting analogous effective results in terms of pain and better results in terms of function. Our study included a knee brace, a lower duration of the ELF application sessions (15 min or 20 min vs. 2 h), and a lower total number of sessions (22 vs. 30). Iannitti and colleagues also applied an analogous treatment period to older patients, which resulted in a reduction in pain and disability [43]. It must be noted that shorter sessions have also been found to be effective.

Physiotherapy and, in particular, instrumental physical therapies in the treatment of KOA are often associated with the use of orthoses to contain pain more quickly and promote better function recovery, but the scientific evidence regarding the use of soft discharge braces attributable to Incred or similar devices is still limited.

Lee et al. [44] showed that knee braces positively affect the patient’s quality of life and, as per NICE guidelines for the conservative management of KOA, should be used in combination with other standard treatments [45]. The use of the Incred knee brace was based on the assumption that embedding germanium into cotton garments could improve the transdermal effect to create a microelectromagnetic field, leading to increased circulation and affecting the inflammatory process [30,44,46]. Additionally, the Incred knee brace has an immobilized elastic sleeve that stabilizes the knee by providing tactile feedback from the skin with a continuous pressure stimulus that can act on pain reduction according to the gate control theory [47,48]. Finally, the use of the knee brace could decrease the biomechanical load on the articulation, which is generally considered a risk factor for KOA progression [49]. In particular, the use of valgus braces without any other addition to the treatment has been found to provide small-to-moderate improvements in terms of pain and function, even with poor compliance during long-term use [50]. The combined use of ELFs and the brace could have the benefit of a quicker effect, diminishing the time of the rehabilitative interventions.

The data from our study would suggest that, in Group 2, patients had a better response in pain resolution, especially at rest (Group 2 vs. Group 1, 1.39 vs. 1.71, *p* = 0.011), and that they probably had a concurrent better response at T3 in functional recovery (for KOOS) compared to Group 1. Considering that the study was conducted using a non-randomized design, these results need further investigation. As reported in the literature [30,42,43], we found a good tolerance to the use of the Incred knee brace as only two patients did not find the brace comfortable. Some studies reported a reduction in pain as early as six weeks after unloading a knee brace in the conservative treatment of KOA [51], compared to usual care, with a positive influence on the dynamics of gait. In fact, in the short-term, knee brace use reflects an increased load on the unaffected limb with a prolongation of the stance phase in both extremities in a long-term effect that persists even after the brace is removed [52,53].

### Strengths and Weaknesses of the Study

This study represents the first attempt at an experiment associated with extremely low-frequency electromagnetic fields (ELF-EMFs) combined with the germanium-embedded knee sleeve in the rehabilitation treatment of patients suffering from KOA.

The dropout of patients remained below 20%, and these interruptions to treatment were mainly for personal and organizational or work reasons. Patients reported no side effects during the sessions or complications, as proof of the safety of the proposed rehabilitative treatment.

Furthermore, comparative studies that consider different rehabilitation protocols with randomized controlled trial (RCT) models would be desirable to contain the selection bias related to the lack of randomization of the sample. A study that considers the inclusion of a third group treated only with the soft knee brace would be particularly desirable. In addition, despite being an observational study, to contain the interview bias, the operator who administered the scales was kept unaware of the patient group, and to ensure adherence to the rehabilitation treatment, two expert physiotherapists were selected who followed the patients from the beginning to the end of the treatment.

## 5. Conclusions

Our results showed that the association of Incred (from 6 h to 8 h a day) with ELF-EMF therapy seems to enhance its positive effect on KOA in short-term pain reduction at rest, with a short follow-up and good maintenance favoring a more rapid recovery of function after six weeks of treatment. Furthermore, in future RCT research developments the combined use of ELF-EMF and other models of soft elastic knee braces should be conducted.

## Figures and Tables

**Figure 1 healthcare-11-01221-f001:**
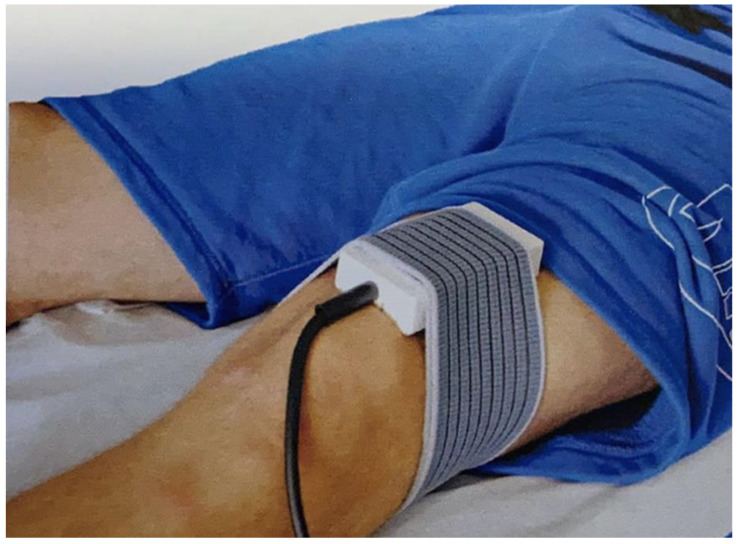
LIMFA application. Representative image of LIMFA application during treatment of knee osteoarthritis.

**Figure 2 healthcare-11-01221-f002:**
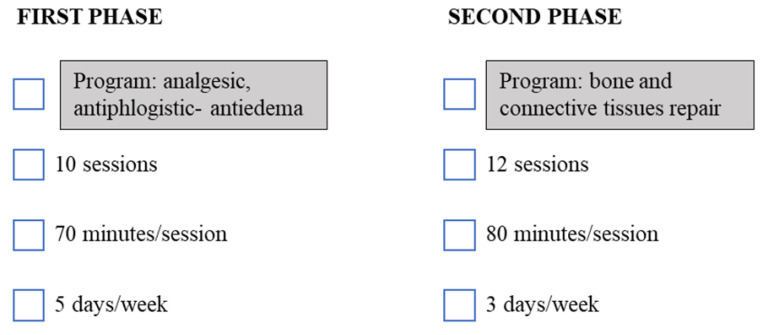
Magnetic Field Protocol. Schematic representation of phases characterizing the magnetic field protocol.

**Figure 3 healthcare-11-01221-f003:**
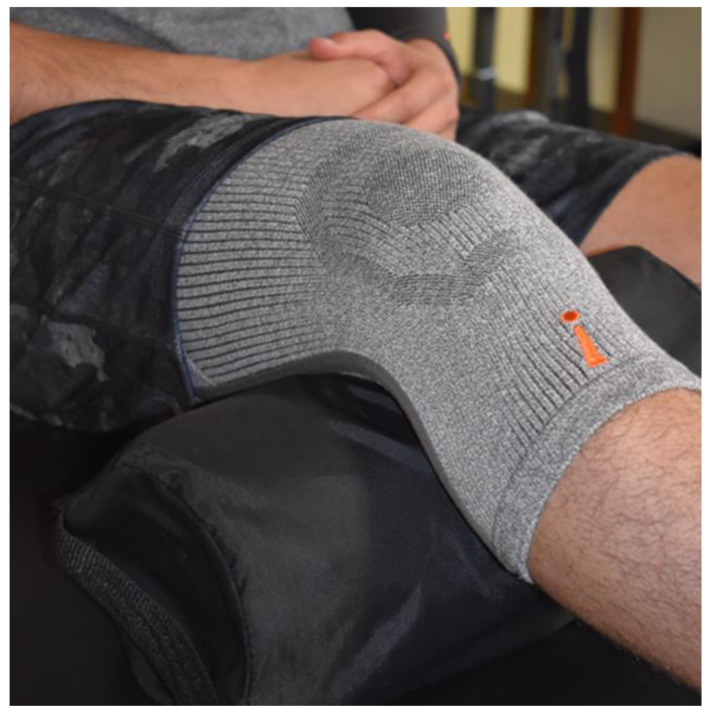
Soft Knee Brace—Incrediwear. Representative image of patient.

**Figure 4 healthcare-11-01221-f004:**
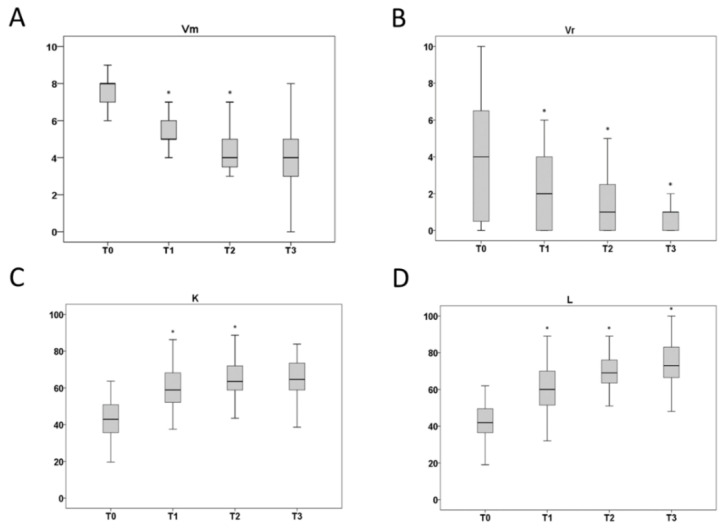
Time-dependent effect of ELF-magnetic field therapy. Measurements of Vm (**A**), Vr (**B**), K (**C**), and L (**D**) in patients submitted to LIMFA therapy were performed at different times (T0–T3). Significance within T0–T3 was obtained by repeated measures ANOVA, considering the therapy as a within factor. Data are expressed as means ± SD (n = 35). * *p* < 0.05 vs. previous ascertained time point. Vm = VAS in motion; Vr = VAS at rest; K= KOOS; Kf = KOOS-functional; L = Ls.

**Table 1 healthcare-11-01221-t001:** Demographic data and clinical disability measures of patients.

Variable	Group 1 (n = 18)	Group 2 (n = 17)	*p*-Value
Gender M/F	5/13	3/14	0.249 ^a^
Age (years)	61.28 ± 7.89	63.76 ± 9.23	0.397 ^b^
BMI	29.16 ± 4.00	26.45 ± 4.22	0.060 ^b^
Time of diagnosis (years)	2.86 ± 2.45	3.65 ± 4.02	0.482 ^b^
KLc	2.83 ± 0.71	2.47 ± 0.62	0.118 ^b^

^a^ Chi-squared test. ^b^ Unpaired Student’s *t*-test. BMI: Body Mass Index; Group 1 = LIMFA; Group 2 = LIMFA + INCRED; KLc = Kellgren–Lawrence classification.

**Table 2 healthcare-11-01221-t002:** Incred effect during ELF-magnetic field therapy. The Vm, Vr, K, Kf, and L measurements in patients submitted to LIMFA therapy without (Group 1) or with (Group 2) INCRED were performed at different times (T0–T3). Significance within T0–T3 was obtained by repeated measures ANOVA, considering the INCRED as a between factor. Data are expressed as means ± SD (n = 25). * indicates a significant overtime difference between groups; # indicates a significant difference between groups in the time point considered. Vm = VAS in motion; Vr = VAS at rest; K= KOOS; L = Ls.

		T_0_		T_1_		T_2_		T_3_		
		Group 1	Group 2	Group 1	Group 2	Group 1	Group 2	Group 1	Group 2	P-Tot
Vm	Mean SD	7.71 1.21	7.78 1.21	5.12 1.27	5.72 1.32	3.65 1.83	4.89 1.71	3.59 1.91	4.44 1.34	0.208
Vr	Mean SD	4.15 2.96	3.63 3.01	1.65 2.06	1.67 2.14	1.41 1.58	1.17 1.58	1.71 0.69	1.39 ^#^ 1.61	0.011 *
K	Mean SD	44.82 10.17	42.19 11.37	63.45 11.68	55.89 10.01	69.02 12.63	58.33 ^#^ 11.73	70.12 12.15	59.42 12.12	0.076
L	Mean SD	43.76 13.92	40.72 11.17	65.47 13.29	56.72 14.64	73.00 12.65	65.89 14.50	77.88 11.69	68.28 15.29	0.492

## Data Availability

The clinical data used to support the findings of this study are available from the corresponding author upon request.

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
