# Peer review of "Combined Rehabilitation Protocol in the Treatment of Osteoarthritis of the Knee: Comparative Study of Extremely Low-Frequency Magnetic Fields and Soft Elastic Knee Brace Effect"

_healthcare, 2023, doi:10.3390/healthcare11091221_

Round 1
Reviewer 1 Report
The topic of this study delivers massive value to EMF therapy for KOA rehabilitation. And the design is overall following a rationale framework. One main problem is language and the quality of figures/tables. The authors should revise following content before it being considered for publication
1. page 2. the short paragraph "Again, ...in this study protocol" needs rewriting to make it free of typo and more readable.
2. The introduction should include more discussion on the mechanism of how soft braces facilitate rehabilitation; some of these are already in the discussion section, which I suggest to move to Introduction.
3. Table 2 needs re-alignment of T0, T1, ...T4. Again, the Mean-SD is not clearly aligned in some columns, please put all SD right below Mean.
4. In discussions supporting Group 2 demonstrate better results over Group 1, use specific number Vr, Vm, K, L to quantitatively show the difference.
Author Response
Reviewer 1
Thank you for the comments. We have revised the MS for English by mother tongue expert and corrected the typo throughout the MS.
1.page 2. the short paragraph "Again, ...in this study protocol" needs rewriting to make it free of typo and more readable.
Thank you. We have revised as follow:
“Furthermore, Ay et al. applied PEMF for 30 minutes, 5 times/week for 3 weeks in KOA with good results with respect to pain reduction and functional recovery [10-11]. Also, Özgüçlü et al. successfully used PEMFs in patients with knee pain using a protocol lasting 30 minutes per session [12-13].”
- The introduction should include more discussion on the mechanism of how soft braces facilitate rehabilitation; some of these are already in the discussion section, which I suggest to move to Introduction.
Thank you for the comment. We have revised in accordance.
- Table 2 needs re-alignment of T0, T1, ...T4. Again, the Mean-SD is not clearly aligned in some columns, please put all SD right below Mean.
Thank you. We have corrected it.
- In discussions supporting Group 2 demonstrate better results over Group 1, use specific number Vr, Vm, K, L to quantitatively show the difference.
In the revised version, we have included the relative values to better show the differences. Thank you.
Reviewer 2 Report
The authors evaluated the efficacy of combined treatment electro- 13 magnetic fields (ELF) and soft elastic knee brace versus ELF using an observational case-control study of knee OA patients. Visual Analogue Scale (VAS), Knee Injury 21 and Osteoarthritis Outcome Score (KOOS), and Lysholm score (Ls). The authors reported that pains at rest (Vr), pain in motion (Vm), KOOS, and Ls were significantly affected by ELF over time. Patients in combination treatment had a better 24 response in pain resolution, and a concurrent better response at T3 in functional recovery. This manuscript is interesting but limited and could use editing for clarity. Comments are listed below.
(1) Abstract: Lacks precise statement of their hypothesis, summary of methods, results, and conclusion.
(2) Introduction: Nice lay out of background information. It would have been informative if the authors compared their proposed combination treatment with the commonly used ELF time/ dose intensity, and other types/ brands of knee brace.
(3) Methods:
a) Assessment techniques used are well established
b) What is the rationale why a third group of knee brace alone not included in the study, their choice of the brace?
c) What is the rationale for the ELF conditions/setting selected?
(4) Results.
a) The number of patients studied is small, would the statistical analysis support level of significance?
b) Are the two groups comparable for statistical analysis?
(5) Discussion-Conclusion
a) What were the limitations compared to the claimed advantages of their method? What is the novelty, added value of their method?
b) The authors concluded the “efficacy” of combined ELF and knee brace, but they used a non-randomized study design with one set of conditions for a short time period. Would this approach be clinically applicable only for these specified conditions and specific type of knee brace?
Overall, the authors’ approach may be interesting but opens further scrutiny.
Author Response
Reviewer 2
The authors evaluated the efficacy of combined treatment electro-magnetic fields (ELF) and soft elastic knee brace versus ELF using an observational case-control study of knee OA patients. Visual Analogue Scale (VAS), Knee Injury and Osteoarthritis Outcome Score (KOOS), and Lysholm score (Ls). The authors reported that pains at rest (Vr), pain in motion (Vm), KOOS, and Ls were significantly affected by ELF over time. Patients in combination treatment had a better response in pain resolution and a concurrent better response at T3 in functional recovery. This manuscript is interesting but limited and could use editing for clarity. Comments are listed below.
Dear Reviewer, in agreement with the first reviewer, we had to further streamline the part regards brace from the introduction and leave it for discussion only.
1.Abstract: Lacks precise statement of their hypothesis, summary of methods, results, and conclusion.
Thank you for the comment. We have revised in accordance.
- Introduction: Nice lay out of background information. It would have been informative if the authors compared their proposed combination treatment with the commonly used ELF time/ dose intensity, and other types/ brands of knee brace.
Thank you for the comment. We have revised.
3.Methods:
a)Assessment techniques used are well established
Thank you.
b)What is the rationale why a third group of knee brace alone not included in the study, their choice of the brace?
Thank you for the comments. Our study is observational, so the lack of a third group represents a limitation. We have discussed it in relative part of discussion section. Furthermore,
we plan to consider including a "knee brace only" group in a future prospective randomized controlled trial.
c)What is the rationale for the ELF conditions/setting selected?
The protocol was set up on the basis of literature studies and other characteristics of the setting of the device used. Thank you for the comment.
4) Results.
- a) The number of patients studied is small, would the statistical analysis support level of significance?
- b) Are the two groups comparable for statistical analysis?
Thank you for the comment. The results are supported by statistical analysis and the tests used are sound.
5) Discussion-Conclusion
- a) What were the limitations compared to the claimed advantages of their method? What is the novelty, added value of their method?
Thanking you. We have added a special section in the discussion: strengths and weaknesses of the study, where these points have been deeply treated.
- b) The authors concluded the “efficacy” of combined ELF and knee brace, but they used a non-randomized study design with one set of conditions for a short time period. Would this approach be clinically applicable only for these specified conditions and specific types of knee brace?
Overall, the authors’ approach may be interesting but opens further scrutiny.
Thanking you. We have added a special section in the discussion: strengths and weaknesses of the study, where these points have been deeply treated.
Round 2
Reviewer 1 Report
Recommend for publication.